# Pro- and Anti-Inflammatory Prostaglandins and Cytokines in Humans: *A Mini Review*

**DOI:** 10.3390/ijms24119647

**Published:** 2023-06-01

**Authors:** Jean-Luc Wautier, Marie-Paule Wautier

**Affiliations:** Faculté de Médecine, Université Denis Diderot Paris Cité, 75013 Paris, France; mpwautier@hotmail.com

**Keywords:** inflammation, prostaglandins, PGE2, TxB2, PGI2, cytokines, IL-1, IL-6, TNFα, TGFβ

## Abstract

Inflammation has been described for two millennia, but cellular aspects and the paradigm involving different mediators have been identified in the recent century. Two main groups of molecules, the prostaglandins (PG) and the cytokines, have been discovered and play a major role in inflammatory processes. The activation of prostaglandins PGE2, PGD2 and PGI2 results in prominent symptoms during cardiovascular and rheumatoid diseases. The balance between pro- and anti-inflammatory compounds is nowadays a challenge for more targeted therapeutic approaches. The first cytokine was described more than a century ago and is now a part of different families of cytokines (38 interleukins), including the IL-1 and IL-6 families and TNF and TGFβ families. Cytokines can perform a dual role, being growth promotors or inhibitors and having pro- and anti-inflammatory properties. The complex interactions between cytokines, vascular cells and immune cells are responsible for dramatic conditions and lead to the concept of cytokine storm observed during sepsis, multi-organ failure and, recently, in some cases of COVID-19 infection. Cytokines such as interferon and hematopoietic growth factor have been used as therapy. Alternatively, the inhibition of cytokine functions has been largely developed using anti-interleukin or anti-TNF monoclonal antibodies in the treatment of sepsis or chronic inflammation.

## 1. Introduction

The discovery of the anti-infectious properties of leukocytes was made in the second decade of the 20th century, as were major discoveries in cellular immunology. After consideration of the role of the bursa of Fabricius cells (BF) in birds, an analogy was made between BF lymphocytes and B lymphocytes, thymus lymphocytes and T lymphocytes in men. The crosstalk between T and B lymphocytes is a new concept relating to immune response. B lymphocytes, like plasmocytes, are the producers of immunoglobulins, which have been chemically characterized [1] as supporting the humoral immunological defense.

The concept of inflammation was previously described in Egyptian hieroglyphs (5000 BC). The cardinal signs were described by Celsus in the first century of CE and later by Galen [2]. The four cardinal signs are rubor (redness), tumor (swelling), dolor (pain) and calor (heat). In the 19th century, according to anatomical and histological observation, the definition was revised. As a new paradigm, it was the subject of controversies between R. Virchow in Germany, J. Cruveilhier in France and, a little later, W. Osler in England and America (Canada and USA). In parallel, a three-hour in vitro observation of a blood clot collected in a glass tube allowed the identification of different conditions. The first was only related to blood coagulation, the second to red blood cell (RBC) sedimentation and the third to the leukocyte concentration. According to the observation, the red blood clot was considered as inflammatory when the red cell blood pack was at the bottom and infectious when it was above the serum level; a third white layer was present which corresponded to increased leukocyte concentration.

Biological tests are routinely performed in hospitalized patients but also in outpatients. Blood cell count, erythrocyte sedimentation rate and C-reactive protein are studied in initial blood tests. Cytokine blood levels of interferon γ, IL-6 and IL-1 are used as markers for infectious diseases such as tuberculosis, while IL-1 and IL-6 are considered to characterize patients with inflammatory disorders, rheumatoid disease and cancer. They may be informative in patients with atherosclerosis and cardiac disorders and can be used to evaluate the degree of severity of some infectious diseases, such as COVID-19 and sepsis [3]. The complexity of the immune response, in addition to the crosstalk between leukocytes, was enriched by the discovery of the cytokines, which have a major role in the modulation of the immune response. The first recognized cytokines were stimulators of cell proliferation and cell signaling. Cytokines are factors which are secreted by different cells, mostly immune cells, and interfere in various functions. The cell proliferation, maturation and activation of several organs perform crucial functions in homeostasis and adaptative reaction to different stresses. Inflammation is an adapted response to physical injury or infection. The mechanisms resulting in inflammation have been widely described, if not completely elucidated, and are associated with an increased concentration of lipid mediators, such as the prostaglandins, leukotrienes, platelet-activating factors and reactive oxygen species mostly secreted by leukocytes, and cytokines and cell adhesion molecule expression [4]. The 20th century was the era of several discoveries which allow better identification of the molecular basis of leukocyte function [5]. In addition, immunoglobulin production regulation and its interaction with the complement system were investigated in the anti-infectious mechanism. These appropriate reactions are at the origin of inflammation and autoimmune diseases.

Decades ago, the resolution of inflammation was considered to be due to the disappearance of the initiators and/or the end of secretion or action of mediators. More recently, the resolution of the inflammatory process has been considered to be an active mechanism following the peak of inflammation.

The paradigm of inflammation has been revisited and the concept of anti-inflammatory drugs reconsidered since they were based on experimental models or inhibition of the mediators involved, such as prostaglandins or cytokines. The successive discoveries of prostaglandins and cytokines have opened up a new area and concept of inflammation.

An overview of prostaglandin and cytokine implication in various diseases from chronic rheumatoid arthritis to COVID-19 disease via atherosclerosis tempted us to try to make a short review on this aspect of inflammation.

## 2. Prostaglandins, Leukotrienes and Eicosanoids

Lipid mediators such as polyunsaturated fatty acids (PUFA) may participate in the initial response to body aggression since they are locally produced [6].

The activation of cyclooxygenase transforms PUFA into prostaglandins and lipoxygenase into leukotrienes (Figure 1). The pro-inflammatory properties of well-identified prostaglandins have been widely described [7].

### 2.1. Cyclooxygenases and Prostaglandins

An increase in COX-2 expression is observed in inflamed tissue. COX-1-derived products initiate the first step of acute inflammation. COX-2 is regulated over the course of hours. COX-2 appears to perform at least two different activities, contributing to the initiation of inflammation and then participating in the resolution of the process [8].

Prostaglandin E2 (PGE2) exhibits various activities and is abundantly produced in the body. PGE2 is involved in the clinical inflammation signs: redness, pain and swelling [9]. In the absence of disease, PGE2 is an important mediator of immune response, blood pressure and intestine function [10]. PGE2 has a contrasting effect in neuroinflammation [11,12]. It enhances pain transmission but limits cytokine and prostaglandin synthesis through EP2 activation [13]. PGE2 is synthetized from PGH2 by cPGES or mPGES-1 [7]. PGE synthase is colocalized with COX-1 in the endoplasmic reticulum [14]. PGE2 binding to cell receptors can modulate macrophages, dendritic cells and T and B lymphocytes in the inflammatory site [15]. 

In the cardiovascular system, prostaglandin I2 (PGI2) participates in the regulation of homeostasis. Vascular endothelial cells and vascular smooth muscle cells are the major productors of PGI2. It has potent platelet antiaggregating properties and vasodilatation activities; it has a local activity and is then converted into 6-keto-PGF1α, which is inactive [16].

Prostaglandin D2 (PGD2) is produced by activated mast cells and is involved in IgE-mediated type 1 acute allergic response [17,18,19,20,21,22]. The opposite pro-inflammatory role of PGD2 may limit allergic reaction through the PDP2 receptor. When PGD2 activates dendritic migration, it limits cytokine production [23]. In atherosclerosis, PGD2 may inhibit the expression of pro-inflammatory genes such as inducible nitric oxide synthase (iNOS) and plasminogen activator inhibitor [24,25]. Thromboxane A2 (TXA2), produced by platelets, induces platelet activation and aggregation but is instable and rapidly converted into inactive TXB2.

Nonsteroidal anti-inflammatory drugs (NSAIDs) are extensively used as anti-pyretic antalgic agents. These compounds inhibit COX-1 and COX-2 and have been classified into several groups: non-complete or complete inhibitors or both (e.g., acetylsalicylic acid, indomethacin, diclofenac, ibuprofen). Acetylsalicylic acid (aspirin) has been widely used in treatment of inflammatory diseases for a long time [26]. Aspirin inhibits COX in a non-reversible way while ibuprofen inhibitory activity is limited in time. Some compounds (meloxicam, celecoxib, nimesulide, etodolac) more specifically block COX-2 [27]. Aspirin is used daily in the treatment of coronary diseases to prevent TXA formation and limit platelet aggregation. On the other hand, it reduces PGI2 formation, which has antiaggregating and vasodilatory activities, which has led to the proposed usage of a low dose of aspirin (less than 100 mg/day) to reduce inhibition of PGI2 formation [28].

### 2.2. Lipoxygenases and Leukotrienes

Leukotrienes (LTs) are generated by lipoxygenases [9]. Leukotriene action is mediated through a specific G-protein-coupled receptor. 

Leukotriene receptors BLT1 and BLT2 are activated by LTB4. LTs are divided into two groups of chemoattractants: LTB4 and CysLTs (LTC4, LTD4 and LTE4). LT involvement in allergic diseases has been widely investigated. In relation to one of the most common chronic allergic diseases, asthma, one 5-LO inhibitor (zileuton) is approved for treatment. In allergic rhinitis (AR), CysLTs are released from inflammatory cells. Atopic dermatitis (AD) pathogenesis is complex; LTB4 and CysLTs may participate by several pathways in the etiopathogenesis. Allergic conjunctivitis occurs through Th2 cell activation. The involvement of CysLTs has been further indicated by the reduction of the oral administration of monteluskast (an anti-asthmatic drug) [29].

The pro-inflammatory lipid mediators known as LTs are involved in immunomodulation but also in the genesis of atherosclerosis [30].

## 3. Cytokines

Cytokines are essential mediators in the inflammatory process (Figure 2). Two cytokines, interleukin-1 (IL-1) and tumor necrosis factor (TNF), which have been recognized for decades, have been shown to coordinate the initiation and cascade of reactions [31]. They participate in the increase in vascular permeability and leukocyte production, stimulation and activation. Instead of being isolated molecules, they are now grouped into families of proteins: the IL-1, IL-6, TNF, TGF and INF families.

### 3.1. IL-1 and IL-6 Families

To try to permit better access to the function of cytokines, a new way of grouping the cytokines into IL-1 and IL-6 families has been proposed in addition to the grouping of other major cytokines, TNF, IFNγ and TGFβ.

The IL-1 family has different ligands with agonist activity (IL-1α, IL-1β, IL-18, IL-33, IL-36α, IL-36β, IL-36γ), receptor antagonists (IL-1Ra, IL-36Ra, IL-38) and an anti-inflammatory cytokine, IL-37 [32]. Several IL-1 family members can be produced by apoptotic cells and may induce sterile inflammation. IL-38 reduces IL-6 production [33] (Table 1).

The IL-6 family of cytokines includes IL-6, IL-11, leukemia inhibitory factor (LIF), oncostatin M (OSM), cardiotropin-1 (CT-1), ciliary inhibitory factor (CNF), neuropoietin (NPN), IL-27 and IL-31. Interleukin-6 (IL-6) is produced by monocytes, endothelial cells and fibroblasts but is also present in mesangial cells, keratinocytes and T and B lymphocytes. Il-6 is a 21–28 kDa glycosylated protein with a four-helix structure. It is involved in inflammation and infection but also in the homeostatic process. As in several homeostatic systems, there is a balance between promoters and inhibitors. The stimulatory activity of IL-6 is mediated by the IL-6 receptor, which consists of two subunits, IL-6R and gp130 (Figure 3). After gp130 dimerization, gp130-associated Jak1 kinase is activated and leads to intracellular transmission, including jak/stat, ERk and P13k [36,37].

Il-6 appears to have a significant effect on immunoglobulins via B cells directly or through the CD4+ T cell helper properties. In patients with systemic lupus erythematosus (SLE), IL-6 potentiates the activity of autoreactive B cells [38,39].

### 3.2. IL-1 and IL-6 Inhibitors

IL-1 and IL-6 are considered as cytokines which play a major role in inflammatory chronic conditions such as rheumatoid arthritis [40], SLE [41] and polyneuritis [42] but also in acute septic shock and COVID-19 infection. This concept is at the origin of the use of monoclonal antibody therapy using antibodies directed against the cytokines or the receptors. 

Neutralization of IL-1β activities results in a rapid reduction in local inflammation and a decrease in inflammatory disease severity. IL-1 receptor antagonist (IL-1Ra) binds to IL-R1 and prevents IL-1α and IL-1β activities [43].

A meta-analysis of the first published trials in patients with COVID-19 treated by anti-IL-6 agents showed no statistically significant reduction in intensive care unit transfer. In other studies, including the RECOVERY study, patients with COVID-19 treated with IL-6 antagonists (e.g., tocilizumab) had a lower mortality rate at 28 days [44].

Interleukin-6 inhibition has been tested in patients who may develop atherosclerosis. A novel anti-IL-6 monoclonal antibody (ziltivekimab) was tested in patients with chronic kidney diseases who were at a high risk of developing atherosclerosis. In the recent RESCUE (Reduction in Inflammation in Patients with Advanced Chronic Renal Disease Utilizing Antibody-Mediated IL-6 Inhibition) trial, a significant reduction in high-sensitivity C-reactive protein (hsCRP) was observed. IL-1 inhibition lowers cardiovascular event rates. Anti-IL-6 strategies can reduce inflammation and atherothrombosis [45].

### 3.3. TNF Family

Tumor necrosis factor is also named cachectin since it produces weight loss in mice. Two cytotoxic factors isolated from macrophages and lymphocytes were named TNF and lymphotoxin, then TNFα and TNFβ. Based on sequence homology, a group of 19 members is considered to belong to the TNF superfamily, having various roles in immunology, inflammation, cell proliferation, angiogenesis and oncogenesis. The TNF members have pro-inflammatory activities, but most of the members have beneficial effects. TNFα stimulates B cell proliferation and differentiation and plays a role in cardiovascular, neurologic, autoimmune and metabolic disorders. The superfamily members are TNFα, TNFβ, lymphotoxin β, CD40L, FasL, CD30L, 4-1BBL, CD27L, OX40L, TNF-related apoptosis-inducing ligand (TRAIL), LIGHT, the receptor activator of NF-κB ligand (RANKL), TNF-related weak inducer of apoptosis (TWEAK), a proliferation-inducing ligand (APRIL), B-cell-activating factor (BAFF), vascular endothelial growth inhibitor (VEGI), ectodysplasin A (EDA), EDA-A1, EDA-A2 and glucocorticoid-induced TNF-related receptor ligand (GITRL) [46].

The members of the TNF family interact with distinct receptors, and the expression of receptors varies between cell types and can transmit a dual signal. TNFα induces several types of signals, including NF-κB, extracellular signal-regulated kinase (ERK), p38 mitogen-activated protein kinase (p38MAPK kinase) and c-Jun N-terminal kinase (JNK) [47].

The pro-inflammatory activity of TNF is dependent upon NF-κB-regulated proteins (IL-6, IL-8, IL-18), iNOS, COX-2 and soluble lectin-like oxidized low-density lipoprotein (sLOX) [48].

### 3.4. TNF Inhibitors

Since TNF family members have been implicated in the pathophysiology of several diseases, they are becoming a target for drug development. Various antagonists are used in the treatment of rheumatoid arthritis, psoriasis, Crohn’s disease and ankylosing spondylarthritis [49].

Several trials targeting the TNF family have been started (phases 1–3) in relation to Crohn’s disease, kidney cancer, colorectal cancer and rheumatoid arthritis. A more complete list was published in [46]. TNF inhibitors have been extensively developed in the last decade and are used in the treatment of inflammatory and autoimmune disorders. Several monoclonal antibodies have been prepared: adalimumab, certolizumab and infliximab. They have been tested in various pathologies and are now currently applied in the treatment of rheumatoid arthritis (RA), Crohn’s disease and amyotrophy lateral sclerosis (ALS). Adalimumab and infliximab are also indicated in the treatment of psoriasis and psoriatic arthritis (PA). Alternatively, etanercept, a receptor derivative (RD), is also an approved treatment for RA, PA and ALS.

### 3.5. Transforming Growth Factor Family

The TGFβ family comprises members which are involved in growth and cell differentiation; they are encoded by 33 genes in mammals [50]. TGFβ is a bifunctional regulator which stimulates or inhibits cell proliferation. The thirty-three known human proteins of the TGFβ family include TGFβ isoforms, activins, nodal and bone morphogenetic proteins (BMPs) and growth and differentiation factors (GDFs). A limited number of receptors named Smad proteins transmit the intracellular signal (Figure 4).

The precursor polypeptides of TGFβ are composed of three segments. The prosegments vary in length (150 to 450 residues). The prosegments are also called latency-associated peptides (LAPs). Before the binding of TGFβ to the cell surface, the LAPs must be released from the mature peptide. TGFβ LAP and β3 LAP have an integrin recognition peptide, Arg–Gly–Asp (RGD), which can bind several integrins, including αvβ6 and αvβ8. TGFα is related to RGF and binds to the EGF receptor. The anti-proliferative activity of TGFβ has been observed in endothelial cells, epithelial cells, hematopoietic and glial cells. TGFβ induces cyclin-dependent kinase inhibitors and inhibits the expression of mediators that contribute to cell proliferation [51]. On the other hand, TGFβ can stimulate cell growth of chondrocytes, osteoblasts, mesenchymal stem cells and fibroblasts [52]. TGFβ controls the cell differentiation of immune cells, blood cells and neuronal cells [53].

### 3.6. Interferon Family

The various biological activities of the different types of interferon have a common ability to induce cell-intrinsic programs. IFNs are either type I or II according to stability in relation to pH. Classification was subsequently based on the specific amino acid sequences and crystal structure [54] and includes IFNα, IFNβ and IFNγ type I and type II. Type I includes 12 IFNαs encoded by different genes, IFNβ, IFNε, IFNκ and IFNω. More recently, a new type III IFN family (IFNλs), which has similarities with the IL-10 family of cytokines, was described [55,56]. Antiviral and anti-tumoral activities and the modulation of the immune response are shared by 21 IFNs [54]. The activities of the three IFN families are initiated by the assembly of the heterodimeric receptor. 

The IFN receptor is followed by activation of the Janus kinase (JAK) intracellular signaling process. JAK phosphorylation activates the STAT-1/STAT-2/interferon regulatory factor 9 (IRF-9) interferon-stimulated gene transcription complex. Each IFN adopts an α helical structure and consists of six secondary structures. The secretion of IFNγ by immune cells (infiltrating lymphocytes (TIL)) upregulates macrophage and dendritic activities. IFNγ exerts an anti-tumoral effect in the tumor vicinity. A major inducer of IFNγ is IL-12, which is present at the site of infection. IL-12 potent activity induces T cell, NK and NKT cell cytolytic activity. In addition to the adapted response of IFNγ, IFN secretion in the host defense, excessive release is associated with chronic inflammation and autoimmune disorders; however, IFN may have a dual role. IFNγ may induce anti-inflammatory molecules such as IL-1 receptor antagonist (IL-1Ra) and IL-18 binding protein, regulating cytokine production and inducing cytokine suppressor signaling [57].

Granulocyte macrophage colony-stimulating factors (GM-CSF) and interleukin-3 (IL-3) amplify the release of IL-1 and TNF. Interleukin-4 (IL-4), interleukin-10 (IL-10), interleukin-13 (IL-13), interferon α (IFNα) and transforming growth factor β (TGFβ) may exhibit anti-inflammatory activities, inducing IL-1 receptor antagonist (IL-1 Ra) and soluble TNF receptor (sTNFR), which limit the activities of IL-1 and TNF. 

## 4. Cytokine Production in Human Pathology

### 4.1. Pathophysiology

The cytokine family and/or receptors have different roles in immune homeostasis and contribute to the pathophysiology of autoimmunity, metabolic and endocrinologic disorders and cardiovascular diseases and cancer.

The pro-inflammatory cytokines IL-1β, IL-6, IL-8, IL-12, IFNγ and TNFα are released at the beginning of the reaction. The cytokines synthetized and secreted by recruited leukocytes are involved according to the stage of reaction and the stimulus: polymorphonuclear cells, macrophages or lymphocytes. The coordinated response to infectious agents which protects the host can become detrimental when chronic. During cancer, proliferation and dissemination of pro-inflammatory cytokines such as IL-1, IFNγ and TNFα may have a deleterious effect. 

Interferon gamma is produced in the host defense against infection and may be elevated during autoimmune and inflammatory diseases. IFNγ can also be involved in anti-inflammatory mechanisms, inducing interleukin-1 receptor antagonist and interleukin-18 binding protein, activating apoptosis [57].

Anti-inflammatory cytokines may inhibit and limit the excess of inflammatory cytokines. The following molecules, IL-1 receptor antagonist, IL-4, IL-6, IL-10, IL-11 and IL-13, are considered to possess anti-inflammatory properties [58]. IL-35 regulated T cell function by suppressing T helper 1 (Th1) and Th17 pathogenic cells in experimental models [59].

### 4.2. Inflammatory Anemia

In inflammatory conditions, inflammatory anemia is sometimes difficult to differentiate from iron deficiency. In inflammatory anemia, hepcidin limits iron access to the bone marrow, reducing erythropoiesis. In mouse models, TNFα, IL-1 and IFNγ suppressed erythropoiesis in vitro. Hepcidin, which is secreted by human hepatocytes, regulates intestinal absorption of iron. IL-1 and IL-6 enhance expression by macrophages, but TNF and IL-6 impair erythropoiesis. Hepcidin binds the iron exporter ferroportin, decreasing delivery from macrophages to erythrocyte precursors. Inhibitors of hepcidin production modulate inflammatory anemia [60]. The first target for the treatment of inflammatory anemia is to treat the underlaying disease; if anemia is severe, blood transfusion or treatment by erythropoietin-stimulating agents may be necessary [61].

### 4.3. Cytokine Storm

Pro-inflammatory and anti-inflammatory cytokines participate in the cytokine storm.

Cytokine storm refers to an influenza-like syndrome in sepsis immunotherapies and may contribute to multi-organ dysfunction. There is no general agreement about the definition of cytokine storm or cytokine release [31]. Nearly all patients have fever, fatigue, headache, anoxia, arthralgia and myalgia. Disseminated intravascular coagulation, dyspnea, vasodilatory shock and acute respiratory distress syndrome (ARDS) can be observed in the most severe cases and can lead to death, despite mechanical ventilation. 

C-reactive protein (CRP) and markers of inflammation are elevated and correlate with the severity [62]. The cytokine levels of IFNγ, IL-6 and IL-10 are found in chimeric antigen receptor T (CART)-cell-induced cytokine storm [63]. Cytokine storm can result from microbial infection. The use of CART cells in the treatment of CD19+ lymphoma can induce cytokine storm. SARS-CoV-2 infection is characterized by heterogenous symptoms ranging from mild fatigue to multi-organ failure. IL-1, IL-6, IP-10, TNF, IFNγ, MIP and VEGF are elevated in COVID-19 patients.

## 5. Hematopoietic Growth Factors

### 5.1. Erythropoietin (EPO)

EPO was initially proposed as a treatment for anemia secondary to renal failure [64]. It is also administrated to reduce the need for transfusion in surgery and to extend the possibility of auto-transfusion. Patients in the preoperative setting, and, more frequently, in the postoperative period, suffer from anemia (10 to 90%). The patient blood cell count and the tests required for a blood transfusion are systematically prescribed in surgical patients. When anemia is found, the cause should be explored, and the cardiac tolerance should be evaluated. When anemia is related to iron, folic acid or vitamin B12 deficiency, the patient should be treated. To reduce blood transfusion, several strategies are applied: first, preoperative treatment, limitation of blood loss during surgery, hemodilution and auto-transfusion. EPO-stimulating agents are administrated to patients at least five days before surgery, associated with intravenous iron injection. This treatment allows the application of pre- or perioperative auto-transfusion, dramatically reducing the risk of infectious agent transmission and alloimmunization [65].

### 5.2. The Leukocyte-Stimulating Factors

The colony-stimulating factors stimulating leukocyte production permit a more rapid recovery from leukoneutropenia secondary to chemotherapy, limiting the risk of infections in patients treated for leukemia or organ cancer [66].

### 5.3. Thrombopoietin

Immune thrombocytopenic purpura (ITP) is first treated by splenectomy, corticotherapy, intravenous immunoglobulins and immunomodulators. Thrombopoietin, after being discovered, was injected in patients suffering from idiopathic thrombocytopenic purpura. Since it is a large molecule which is difficult to synthetize, active peptides, or, more recently, chemical agents, have replaced thrombopoietin for the treatment [67]. Recombinant thrombopoietin and thrombopoietin receptor agonist (TPO-RA) have become a second line of treatment. Romiplostin and eltrombopag, two TPO-RAs, have been tested in patients with ITP. The two TPO-Ras bind to TPO receptor and activate megakaryocyte progenitor, which leads to platelet production [68,69].

## 6. Conclusions

After being clarified, the different functions of prostaglandins are still only partly delineated. Despite the frequent use of NSAIDs for the treatment of inflammatory diseases, the side effects of several compounds, including those of aspirin, mean that the research of new drugs is a major goal for the pharmaceutical industry. In addition to the most popular cytokines, IL-1, TNF and IL-6, the roles of the recently discovered cytokines and receptors have been explored in various pathologies, and the complexity in immune regulation remains incompletely understood. The development of monoclonal antibodies which can inhibit different cytokines is encouraging for the treatment of chronic inflammatory diseases, but the dual function of some cytokines, which possess pro- and anti-inflammatory activities, exhibits the difficulty in the approach blocking the function. Different investigations demonstrating the infiltration of tumors by Treg cells raised the important question of the function of autonomous or non-autonomous cell reaction. The participation of cytokines, not only in immunopathology but also in degenerative diseases of the central nervous system, cardiovascular disorders and cancer, has extended the potential therapeutic aspect of cytokine modulators. It was reported in the CANTOS study that blockers of IL-1 may be beneficial against atherosclerosis, lung cancer and arthrosis [70].

A better understanding of the role of the different cytokine family members and genetic control may offer a new strategy for the development of new or already known therapeutic tools. The historical concept of inflammation has been revitalized with the discovery of inflammatory lipids (mostly prostaglandins) and cytokines, a new method of communication between cells.

## Figures and Tables

**Figure 1 ijms-24-09647-f001:**
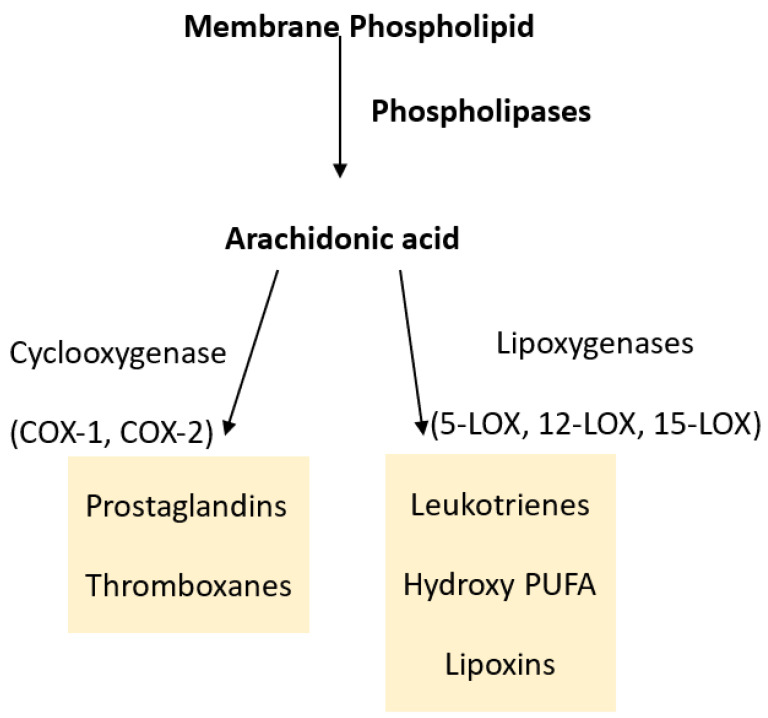
Metabolism of arachidonic acid by different pathways resulting in lipid mediators of inflammation.

**Figure 2 ijms-24-09647-f002:**
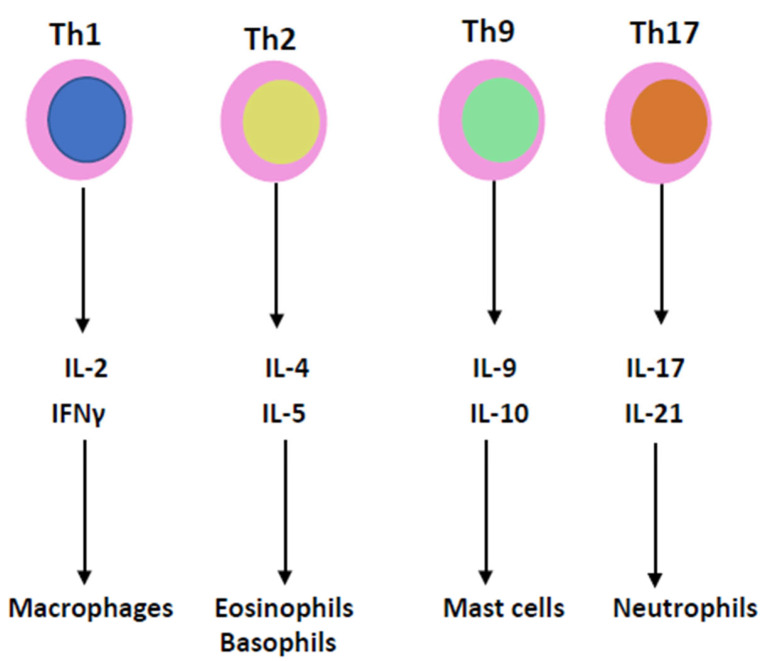
T lymphocyte helpers of T cell subgroups 1, 2, 9 and 17 produce IL-2, IL-4, IL-9, IL-17, IFNγ, IL-5, IL-10 and IL-21, which are involved in the pathophysiology of the cytokine storm.

**Figure 3 ijms-24-09647-f003:**
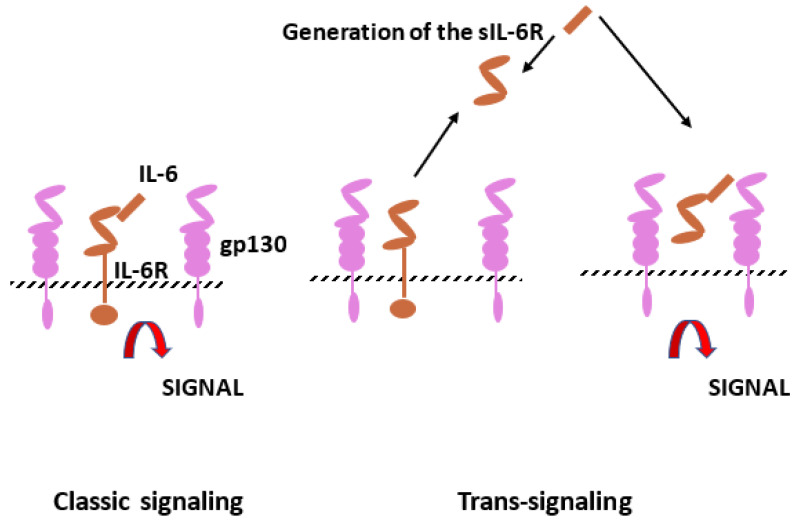
Interleukin-6 (IL-6) signaling pathways. Il-6 can bind to IL-6 membrane bound receptor (IL-6R), which corresponds to the classic signaling but also to soluble IL-6R (sIL-6R) resulting from limited proteolysis by ADAM proteases. Cells with gp130 can be stimulated by the complex IL-6 and sIL-6R (trans-signaling).

**Figure 4 ijms-24-09647-f004:**
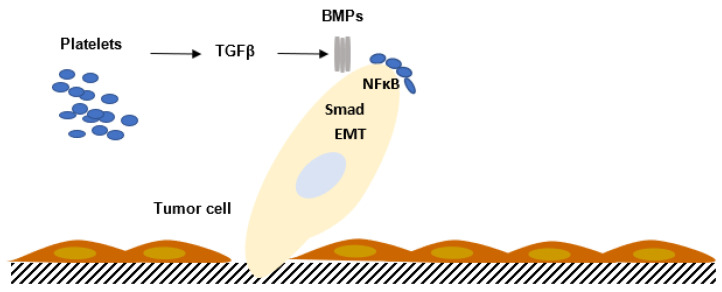
Interaction between platelets and tumor cells activates the TGFβ/SMAD pathway in tumor cells, enhancing tumor cell migration and increasing the metastatic process. The contact between platelets and tumor cells, via TGFβ/SMAD and NFκB, potentiates epithelial–mesenchymal transition (EMT) and metastasis. BMPs: bone morphogenetic proteins.

**Table 1 ijms-24-09647-t001:** Interleukin-1 and interleukin-6 families.

**Interleukin-1 (IL-1) Family** [34,35]			
Cytokines	Pro-Infl	Anti-Infl	Cell Source
IL-1α	**+**		Epithelial, endothelial, stromal cells
IL-1β	**+**		Monocytes/macrophages
IL-18	**+**		Kupffer cells, intestinal epithelial cells
IL-33	**+**		Keratinocytes, endothelial cells, epithelial cells, fibroblast-like cells
IL-36α	**+**		Epithelial cells
IL-36β	**+**		Epithelial cells
IL-36γ	**+**		Epithelial cells
**Interleukin-6 (IL-6) Family** [36]			
Cytokines	Pro-Infl	Anti-Infl	Cell Source
IL-6	**+**	**+**	B cells, T cells, monocytes, fibroblasts, endothelial cells
IL-11		+	Stroma cells, fibroblasts
CNF	**+**	**+**	Astrocytes
LIF	**+**	**+**	Pleiotropic
OSM	**+**	**+**	Pleiotropic
CT-1	**+**		Myocytes
CLC	**+**	**+**	Lymphocytes
IL-27		**+**	T cells
IL-31	**+**		Monocytes/macrophages, dendritic cells, T cells

Members of the Il-1 and Il-6 families perform pro- and anti-inflammatory activities, which are summarized on Table 1, as is cell origin. Abbreviations used in the table: CNF: ciliary neurotrophic factor, LIF: leukemia inhibitory factor, OSM: oncostatin M, CT-1: cardiotrophin 1, CLC: cardiotrophin-like protein. Pro-Infl means pro-inflammatory, and Anti-Infl means anti-inflammatory.

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
