# Peer review of "Pro- and Anti-Inflammatory Prostaglandins and Cytokines in Humans: A Mini Review"

_ijms, 2023, doi:10.3390/ijms24119647_

Round 1

Reviewer 1 Report

The paper has too many drawbacks. 

1. The abstract is chaotic and without logical connection between the sentences. This is also valid for the whole manuscript. There is no idea behind the paper, no aim. The title is misleading.

2. Section 1 on prostaglandins doesn`t add anything new to the topic, as the following sections.

3. The material is written more as an chapter in student book than a paper.

4. Sections 3.2 and 3.4 come out of the blue

5. The conclusion even add information, that was not mentioned before.

Author Response

Reviewer 1 report (Round 1)

We thank reviewer 1 to offer the opportunity to explain what was our goal and potentially to improve our article. Working in a University Hospital for decades we have collaborated with different specialists. Clinicians and surgeons are not familiar with the molecular aspects of the medicine and are sometimes looking for a comprehensible review article.

Reviewer 1 has a clever way of writing criticism which can be applied whatever the topic is. In addition to chaotic and inorganized, stochastic can be an alternative. It may also be added that some important references have been omitted, which is frequent.  

The paper has too many drawbacks. 

  1. The abstract is chaotic and without logical connection between the sentences. This is also valid for the whole manuscript. There is no idea behind the paper, no aim. The title is misleading.

The title has been changed

  1. Section 1 on prostaglandins doesn`t add anything new to the topic, as the following sections.

We don’t agree

  1. The material is written more as an chapter in student book than a paper.

There is no material section

  1. Sections 3.2 and 3.4 come out of the blue

There is no section 3.4

  1. The conclusion even add information, that was not mentioned before.

There are different ways to write a conclusion according to the subject and the style of the authors

Reviewer 2 Report

This review is described about the inflammatory control by prostaglandins and cytokines. It is concisely summarized about each mediator. However, I think it should be more information. Moreover, although the title is “A new balance between the pro and anti-inflammatory prostaglandins and cytokines”, I cannot find what is “New”. In addition, in Fig. 1, leukotrienes are included in the figure, but it was not mentioned, although leukotrienes are important in the regulation of inflammation. There is also a concern below, that should be addressed.

Line 105: The sentence “PGE2 is colocalized with COX-1 in the endoplasmic reticulum,” is strange, because PGE2, a lipid is not colocalized with enzyme COX-1. “PGE synthase is co-localized with COX-1 in the endoplasmic reticulum” is better, if this is correct.

Author Response

Reviewer 2 report (round 1)

The authors are extremely grateful to reviewer 2 for the review report which is precise and help us to improve the manuscript.

This review is described about the inflammatory control by prostaglandins and cytokines. It is concisely summarized about each mediator. However, I think it should be more information. Moreover, although the title is “A new balance between the pro and anti-inflammatory prostaglandins and cytokines”, I cannot find what is “New”. In addition, in Fig. 1, leukotrienes are included in the figure, but it was not mentioned, although leukotrienes are important in the regulation of inflammation. There is also a concern below, that should be addressed.

According to the recommendation of the reviewer 2, we changed the title.

A new section dealing with leukotrienes has been added in the prostaglandin part

Line 105: The sentence “PGE2 is colocalized with COX-1 in the endoplasmic reticulum,” is strange, because PGE2, a lipid is not colocalized with enzyme COX-1. “PGE synthase is co-localized with COX-1 in the endoplasmic reticulum” is better, if this is correct.

The sentence has been changed to that of the reviewer

Reviewer 3 Report

This review is concerned with two types of inflammatory mediators: lipid mediators and cytokines. Each is a broad area of research. This review may help the non-specialist reader to get a broad picture of them. I have the following concerns.

1, The title of the article uses the term "New balance”. Althoug this review describes inflammatory and anti-inflammatory properties of each mediator, there is little description about new balance. If the authors stick to this title, they should make it clear what is new balance. Otherwise, change the title.

2The following points need to be corrected.

Line 147-150 This paragraph is not related to the preceding and following paragraphs

Table1 Arrangement in the table is broken.

Line 157 Il-1 Il-6

Line 158 Different abbreviations  Different from what?

Figure 3 It is unclear what the illustration in the figure means.

Figure 3 The contents of the text and the figure legend are not consistent.

Line 186 IL-2R2 ?

Line 186 γc is IL-2Rγc

Line 186 Il-2

Line 187 IL-2β ?

Line 187 stimulated is in the past tense, and there are other parts where the past tense is mixed up.

Line 188 What is STATB?

Line 208 What is GITRL?

Line 209 - 210 the expression varies  expression of receptors?

Line 210 What does "bipolar signals" mean?

Line 215 What is SLOX?

Line 218 "," should be "."

Figure 4: The text and the figure are not consistent with each other.

Figure 4 It is unclear what it means that monocytes release IL-1beta and TNFα.

Line 243 IFNalfa/beta and IFNgamma type I and type II.

Line 279 Interferon-gamma (IFNgamma) is a repetition.

Line 280 Intereleukin (IL) is a repetition

Line 291-293 I don't understand the sentence structure

Line 307 What is CART cell

Table 2 The place of RD should be corrected

Line 335 "." Line 335 "." should be inserted.

Line 353 against should be agonist

Line 361-364 Unclear sentence.

Author Response

Reviewer 3 report (Round 1)

This review is concerned with two types of inflammatory mediators: lipid mediators and cytokines. Each is a broad area of research. This review may help the non-specialist reader to get a broad picture of them. I have the following concerns.

1, The title of the article uses the term "New balance”. Althoug this review describes inflammatory and anti-inflammatory properties of each mediator, there is little description about new balance. If the authors stick to this title, they should make it clear what is new balance. Otherwise, change the title.

 According to the recommendation of the reviewer 3, we changed the title.

2、The following points need to be corrected.

Line 147-150 This paragraph is not related to the preceding and following paragraphs

This paragraph has been deleted

Table1 Arrangement in the table is broken.

We have to recompose the table 1 which is now new

Line 157 Il-1 Il-6

Line 158 Different abbreviations  Different from what?

This section has been completely remodeled

Figure 3 It is unclear what the illustration in the figure means.

Figure 3 The contents of the text and the figure legend are not consistent.

The figure 3 has been completely redesigned and is now a new one

Line 186 IL-2R2 ?

Line 186 γc is IL-2Rγc

Line 186 Il-2

Line 187 IL-2β ?

Line 187 stimulated is in the past tense, and there are other parts where the past tense is mixed up.

Line 188 What is STATB?

This part has been deleted

Line 208 What is GITRL?

The meaning of the abbreviation is given

Line 209 - 210 the expression varies  expression of receptors?

Line 210 What does "bipolar signals" mean?

The sentence has been changed

Line 215 What is SLOX?

The meaning of the abbreviation is given

Line 218 "," should be "."

“should be” has been added

Figure 4: The text and the figure are not consistent with each other.

Figure 4 It is unclear what it means that monocytes release IL-1beta and TNFα.

The figure 4 and the legend have been changed

Line 243 IFNalfa/beta and IFNgamma type I and type II.

The spelling has been changed

Line 279 Interferon-gamma (IFNgamma) is a repetition. corrected

Line 280 Intereleukin (IL) is a repetition. corrected

Line 291-293 I don't understand the sentence structure. The sentence has been deleted

Line 307 What is CART cell

The meaning of the abbreviation is given

Table 2 The place of RD should be corrected

Corrected

Line 335 "." Line 335 "." should be inserted. Done

Line 353 against should be agonist. Corrected

Line 361-364 Unclear sentence. Deleted

Round 2

Reviewer 1 Report

I have been carefully reviewed your revised article with the id number " ijms-2324107 ". In my opinion, this revised article incorporates all of the points raised in the original draft to the best of my knowledge.
The data is presented in an appropriate way and all of them are discussed from different angles.
The article is written very detailed. Figures and tables have been added to better visualize the information.
The reader understands the importance of the topic although the large volume of scientific data. All refinements made the article more interesting for readers from multiple backgrounds.
Best wishes to all of the authors who contributed to the production of this work and congratulations on their future endeavors.

Author Response

We thank reviewer 1 for the nice words and the encouragements.

Reviewer 2 Report

Although the changed title is so big, there is too little content. This review is not so informative. Moreover, the sub-heading is so strange, because 1-1 and 1-2 are enzymes, but 1-3 is chemicals. In addition, leukotrienes are not involved in the prostaglandins. They are eicosanoid. Anyway, I think this review is not informative to the readers.

Author Response

The title has been changed on the request of two reviewers. The title of a review is broader than that of an original article but we don’t understand what is a “big” title. Is it an “overstatement, a “too long” title?  A possible new title is proposed.

The section dedicated to prostaglandins is limited compared to that dealing with cytokines. The managing editor made the proposal that if accepted the article will be published in the special issue : cytokines.  Most of the new information is in the cytokine part introducing the concept of IL-1 IL-6 families and the anti-inflammatory properties of IL-6. It did not appear fair to us to forget the prostaglandins in this review. To stick to the reviewer recommendation subtitle has been modified (line 81)

About the word not informative (twice) in the comments it may be understood in different ways. The other reviewers have a different comment.

This short review was not written for experts in the field but to propose a readable article for non-specialists to point out that prostaglandins and cytokines , beside the inflammatory activities, could have anti-inflammatory properties.

The number and the quality of references demonstrate that the article is well documented and consequently should provide some information.

Reviewer 3 Report

The manuscript has been properly revised. It is now suitable for publication.

Author Response

We thank reviewer 3 for help in improving the article